# Detecting Data Contamination from Reinforcement Learning Post-training for Large Language Models

**Yongding Tao**[1]   **Tian Wang**[1]   **Yihong Dong**[1]   **Huanyu Liu**[1]   **Kechi Zhang**[1]
**Xiaolong Hu**[2]   **Ge Li**[1]
[1] School of Computer Science, Peking University    [2] New H3C Technologies Co., Ltd
ydtao25@stu.pku.edu.cn   lige@pku.edu.cn

## Abstract

Data contamination poses a significant threat to the reliable evaluation of Large Language Models (LLMs). This issue arises when benchmark samples may inadvertently appear in training sets, compromising the validity of reported performance. While detection methods have been developed for the pre-training and Supervised Fine-Tuning stages, a critical research gap exists for the increasingly significant phase of Reinforcement Learning (RL) post-training. As RL post-training becomes pivotal for advancing LLM reasoning, the absence of specialized contamination detection methods in this paradigm presents a critical vulnerability. To address this, we conduct the first systematic study of data detection within RL post-training scenario and propose **Self-Critique**. Our method is motivated by a key observation: after RL phase, the output entropy distribution of LLMs tends to collapse into highly specific and sparse modes. Self-Critique probes for the underlying policy collapse, i.e., the model's convergence to a narrow reasoning path, which causes this entropy reduction. To facilitate this research, we also introduce **RL-MIA**, a benchmark constructed to simulate this specific contamination scenario. Extensive experiments show that Self-Critique significantly outperforms baseline methods across multiple models and contamination tasks, achieving an AUC improvement of up to **30%**. Whereas existing methods are close to a random guess for RL-phase contamination, our method makes detection possible. Our benchmark and code are available at https://github.com/yongding-tao/RL-Data-Contamination.

## 1 Introduction

The reliability of Large Language Model (LLM) evaluations is seriously threatened by data contamination. This happens when benchmark test samples accidentally get included in the training data, which can invalidate the model's reported performance. To solve this problem, many researchers have developed detection methods, but they have almost exclusively focused on the pre-training and Supervised Fine-Tuning (SFT) (Dong et al., 2024; Fu et al., 2024; Shi et al., 2024; Zhang et al., 2024b; Xie et al., 2024b; Zhang et al., 2025a;b) stages. However, these efforts have left a major gap: the increasingly important phase of Reinforcement Learning (RL) post-training. We believe this is a critical oversight, because powerful techniques like Reinforcement Learning with Verifiable Rewards (RLVR) (Shao et al., 2024a; Guo et al., 2025; Yu et al., 2025) are now essential for improving LLM reasoning. This makes the RL stage a major potential source of contamination that has been largely overlooked.

The challenge of detecting RL-phase contamination stems from a fundamental shift in the training objective, rendering existing methods ineffective. Both pre-training and SFT are likelihood-based paradigms; they train models to maximize the probability of observed data. This process naturally creates strong, likelihood-based signals, such as unusually low perplexity, which most current detectors are built to identify. By contrast, RL, especially RLVR, operates on a reward-maximization principle. The policy is not trained to mimic a ground-truth distribution but is instead guided by

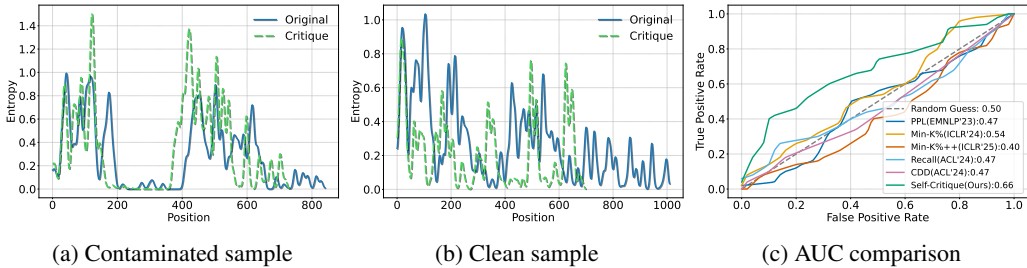

(a) Contaminated sample      (b) Clean sample      (c) AUC comparison

Figure 1: Motivation behind **Self-Critique**. After RL post-training, entropy distributions become sparse. (a) For contaminated samples, the critique reasoning path remains highly similar to the original one, indicating policy collapse and memorization. (b) Clean samples exhibit greater divergence between the original and critique reasoning paths. (c) Our method achieves a significantly higher AUC while existing baselines perform close to random guess.

sparse reward signals to find a successful reasoning path. This approach often enables stronger generalization than SFT (Kirk et al., 2024; Chu et al., 2025), but by decoupling from likelihood-based objectives, it also erases the very signals that traditional detectors rely on. Consequently, RL-phase contamination becomes a uniquely challenging problem, creating an urgent need for a new class of detection methods specifically designed for this reward-driven setting.

Given that likelihood-based signals are ineffective, our search for a new detection method begins with identifying a signal inherent to the reward-driven training process. Recent studies on RL's training dynamics point to a promising candidate: the phenomenon of **policy collapse**. Specifically, RL narrows the search space to improve pass@1 accuracy, often at the cost of lower pass@k performance (Havrilla et al., 2024; Shao et al., 2024b; Yue et al., 2025; Dong et al., 2025), and produces distinctive entropy patterns, such as high-entropy concentration on certain tokens (Wang et al., 2025a; Cheng et al., 2025; Song et al., 2025). These findings suggest that entropy could serve as a powerful indicator of this collapse and its associated path dependency. However, our initial investigations revealed that using entropy directly as a contamination signal is unreliable. The reason is that policy collapse is a general behavior of RL and can occur even on clean samples not seen during training. As shown in Figures 1a and 1b, both contaminated and clean samples can exhibit sparse token-level entropy. This implies that a simple passive check is insufficient. Therefore, we introduce an active **probing mechanism** to expose the underlying differences. We find that when the model is asked to **generate an alternative reasoning path given its initial response** (*self-critique*), contaminated samples struggle to deviate, resulting in highly similar entropy curves (Figure 1a). In contrast, the model shows greater flexibility on clean samples, leading to more distinct entropy patterns (Figure 1b).[1]

Building on these observations, we introduce **Self-Critique**, an entropy-based detection method that applies our self-critique probing strategy. The core idea is to instruct the model to generate two distinct responses for the same problem; samples where the two responses exhibit high similarity in their entropy space are flagged as contaminated. A detailed workflow is shown in Figure 2. However, rigorously evaluating this method is challenging, as no existing benchmark can isolate and simulate contamination purely within the RL phase. To overcome this hurdle, we also developed **RL-MIA** (Reinforcement Learning Membership Inference Attack), a new benchmark constructed for this specific purpose. Using RL-MIA across challenging math and logic datasets, we show that Self-Critique is highly effective. As previewed in Figure 1c, our method significantly outperforms existing detectors, which operate near the level of random guess.

Our main contributions are summarized as follows:

❶ To the best of our knowledge, we present *the first systematic study of data contamination detection in the RL post-training phase of LLMs*, highlighting a critical yet overlooked problem.

❷ We propose **Self-Critique**, an entropy-based detector that measures RL-induced policy collapse via self-critique probing. Across four tasks and multiple models, Self-Critique

---

[1] We also provide visualizations of the contamination score distribution in Appendix D.

consistently outperforms baselines, which perform near random guess, achieving an AUC improvement of up to 30%.

❸ We introduce **RL-MIA**, a new benchmark that simulates RL-specific contamination scenarios across math and logic tasks, enabling the rigorous evaluation of detection methods.

## 2 RELATED WORKS

In this section, we outline the two most relevant directions and associated papers of this work.

**Data Contamination Detection**    Data contamination detection can be regarded as a specific instance of membership inference attacks (MIA), which were initially introduced to measure memorization and privacy risks (Shokri et al., 2017; Carlini et al., 2019; Mireshghallah et al., 2022a). Recently, the issue of data contamination in LLMs has drawn increasing attention, as it directly undermines the validity of benchmark evaluations (Sainz et al., 2023; Xu et al., 2024a;b; Wu et al., 2025). Prior work on data contamination detection in LLMs has mainly focused on the pre-training and Supervised Fine-Tuning stages (Mireshghallah et al., 2022b; Fu et al., 2023; Mattern et al., 2023; Shi et al., 2024; Xie et al., 2024b; Gonen et al., 2023; Dong et al., 2024; Zhang et al., 2025b). In these stages, models largely rely on memorizing training data for learning (Zeng et al., 2024a; Chu et al., 2025; Wang et al., 2025b), a process that naturally creates strong, likelihood-based signals—such as unusually low perplexity—that most current detectors are built to identify. In contrast, during the RL post-training phase, LLMs are optimized to autonomously explore reasoning trajectories. This reward-driven objective decouples the model's behavior from simple likelihood metrics, posing a unique challenge for conventional detection methods.

**Entropy in Reinforcement Learning Post-training**    Reinforcement learning has become a crucial paradigm for post-training large language models. Leveraging reinforcement learning with verifiable rewards substantially enhances LLM's reasoning capabilities (Jaech et al., 2024; Guo et al., 2025). A key factor in RL post-training is entropy: high entropy promotes exploration via stochastic policies, while low entropy favors exploitation through deterministic behavior. A common challenge in RL post-training is entropy collapse (Cui et al., 2025; Liang et al., 2025), where policy entropy decreases dramatically in the early stages of training, leading to premature convergence and restricted exploration. To address this, entropy management strategies regularize entropy to prevent rapid collapse (O'Donoghue et al., 2016; He et al., 2025; Wang et al., 2025c) or use high-entropy signals to encourage inherently exploratory reasoning behaviors (Cheng et al., 2025; Vanlioglu, 2025; Tan & Pan, 2025), thus maintaining a balance between exploration and exploitation. In reasoning tasks, high-entropy tokens indicate uncertain decision points and are assigned stronger RL updates, while low-entropy tokens, which correspond to more deterministic outputs, receive smaller updates (Li et al., 2025; Wang et al., 2025a; Tang et al., 2025). As a result, the trained model develops distinct entropy patterns across tokens. In this work, we analyze these entropy patterns before and after self-critique to detect potential data contamination.

## 3 THE CHALLENGE OF CONTAMINATION DETECTION IN RL

In this section, we formalize the problem of detecting data contamination in the RL post-training phase of LLMs, and then highlight why detection methods based on likelihood, which are effective in pre-training and SFT, become unreliable in RL. Finally, we introduce token-level entropy as a lens to analyze RL-induced policy collapse, which lays the foundation for our proposed method.

### 3.1 PROBLEM DEFINITION

We consider the task of detecting data contamination in the RL post-training phase of Large Language Models. Formally, this can be framed as a Membership Inference Attack(MIA) problem: given a model $\mathcal{M}$ that has undergone RL post-training and a sample $x$, the goal is to determine whether $x$ was included in the RL training dataset $D_{\text{RL}}$. A detector is a function $f(\mathcal{M}, x) \rightarrow \{0, 1\}$, where 1 indicates membership (contamination) and 0 indicates non-membership. We focus on the black-box setting (Shi et al., 2024), where the detector can only query $\mathcal{M}$ for outputs, without access to internal states, gradients, or training data.

## 3.2 WHY RL POST-TRAINING IS A UNIQUE CASE

Existing data detection methods are largely designed for training paradigms whose objectives are rooted in Maximum Likelihood Estimation (MLE). However, the objective of RL post-training is fundamentally different, which introduces unique and significant challenges for detection.

Both pre-training and Supervised Fine-Tuning are governed by an MLE-based objective. Their goal is to train a model $\mathcal{M}$ with parameters $\theta$ to maximize the likelihood of the observed data by minimizing the negative log-likelihood loss.

For **pre-training**, the model learns from a vast corpus of unlabeled text $D_{\text{pretrain}}$. The objective is next-token prediction, aiming to learn a general distribution of the language. For a text sequence $x = (x_1, x_2, \ldots, x_T)$, the loss is:

$$\mathcal{L}_{\text{Pretrain}}(\theta) = - \sum_{x \in D_{\text{pretrain}}} \sum_{t=1}^{T} \log p_\theta(x_t | x_{<t}) \tag{1}$$

For **SFT**, the model learns from a dataset of prompt-response pairs $D_{\text{SFT}} = \{(q, r)\}$. The objective is to learn to follow instructions and generate helpful responses. The model is trained to maximize the likelihood of the target response $r = (r_1, \ldots, r_K)$ given the prompt $q$:

$$\mathcal{L}_{\text{SFT}}(\theta) = - \sum_{(q,r) \in D_{\text{SFT}}} \sum_{t=1}^{K} \log p_\theta(r_t | q, r_{<t}) \tag{2}$$

Despite their different data sources, both paradigms (Eq. 1 and 2) share the same underlying principle: they **directly train the model to assign high probabilities to sequences seen in the training data**. This provides a clear signal for detection methods like Perplexity (Gonen et al., 2023) and Min-K% Prob (Shi et al., 2024), which are built upon this likelihood principle.

In stark contrast, **RL post-training** (and specifically RLVR) does not directly optimize for likelihood. Its objective is to update the policy $\pi_\theta$ to maximize the expected *reward* $\mathcal{R}$ from a set of generated outputs $\{o_i\}$ given a prompt $q$. The objective for a method like GRPO (Shao et al., 2024a) can be abstracted as:

$$\mathcal{J}_{\text{RL}}(\theta) = \mathbb{E}_{q \sim D_{\text{RL}}, \{o_i\} \sim \pi_{\theta_{\text{old}}}} \left[ f(\mathcal{R}(o_i), \pi_\theta) \right], \tag{3}$$

where $f(\cdot)$ is a function of the reward, the current policy, and a reference policy. The key distinction is that the optimization is driven by an external, often sparse, reward signal $\mathcal{R}(o_i)$ (e.g., 1 for a correct final answer, 0 otherwise), not by the token-level log-probabilities of the ground-truth response. This decouples the model's final behavior from simple likelihood metrics, rendering many existing detection approaches that rely on this signal ineffective.

## 3.3 ENTROPY AS A NEW SIGNAL FOR RL DETECTION

Recent studies (Yue et al., 2025; Wang et al., 2025a; Cui et al., 2025) have shown that RL post-training frequently leads to *policy collapse*: for samples that receive consistent reward, the model converges to a narrow reasoning path, producing overly stable outputs . This phenomenon is reflected in the *token-level entropy*. For each decoding step $t$, the token-level entropy is

$$H_t = - \sum_{v \in V} p_\theta(v \mid x_{<t}) \log p_\theta(v \mid x_{<t}), \tag{4}$$

and the entropy sequence $E = \{H_t\}_{t=1}^{T}$ measures uncertainty along the generated trajectory. Empirical observations show that RL tends to push entropy sequences into sparse patterns, where many tokens are nearly deterministic. Crucially, this collapse is stronger for contaminated samples that were explicitly rewarded during RL training, whereas clean samples retain more variability when probed.

These insights suggest that contamination detection in RL requires moving beyond likelihood and instead measuring the policy's dependence on specific reasoning paths. Token-level entropy provides a natural signal for this purpose, which directly motivates our **Self-Critique** method: by asking the model to regenerate an alternative reasoning path conditioned on its initial response and comparing the entropy sequences, we can reveal whether a sample was memorized during RL training.

# 4 DETECTION VIA SELF-CRITIQUE

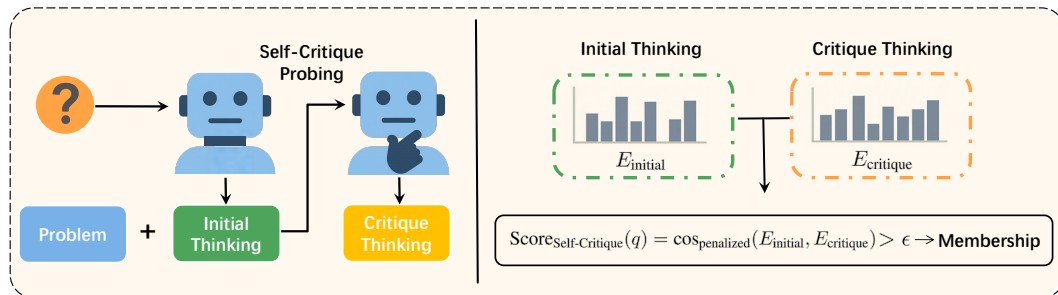

Figure 2: Overview of the **Self-Critique** detection workflow. The method compares token-level entropy sequences between the initial response and the self-critique response. High similarity in entropy space indicates contamination (policy collapse), while low similarity indicates clean samples.

Our method is motivated by the hypothesis that RL post-training induces high-reward path dependence for contaminated samples. Concretely, for a problem $q$ seen during RL training, the policy $\pi_\theta$ tends to converge to a highly rewarded and thus similar response trajectory. In contrast, for problems not seen during RL training, the model is more likely to produce an alternative reasoning path when prompted.

The **Self-Critique** method quantifies this dependency. We first elicit the model's most confident (deterministic) response, and then ask the model to produce a different solution *conditioned on* the initial response. We compare the token-level entropy sequences of the two generations to measure the degree of path dependence. An overview is shown in Figure 2.

## 4.1 THE SELF-CRITIQUE DETECTION PROCESS

Let $\mathcal{M}$ be a large language model with parameters $\theta$, and let $q$ be the problem under test. We use a deterministic decoding strategy (e.g., greedy decoding) to obtain the model's most confident response as the reference.

**Step 1: Initial response.** We construct the initial prompt $P_1$ by embedding $q$ into a chat template $T$, and obtain the model's response:

$$r_1 = \mathcal{M}(T(q)). \tag{5}$$

We then compute the token-level entropy sequence for this response, $E_1 = \{H_t(r_1)\}_{t=1}^{|r_1|}$, which serves as the baseline reasoning trajectory.

**Step 2: Self-critique response.** We form a self-critique prompt $P_2$ by augmenting $q$ with an instructional meta-prompt $I_{\text{critique}}$[2] and the text of $r_1$:

$$q' = q \oplus I_{\text{critique}}(r_1), \tag{6}$$

where $\oplus$ denotes appending to the user content within the prompt structure. We then generate the second response and its entropy sequence:

$$r_2 = \mathcal{M}(T(q')), \quad E_2 = \{H_t(r_2)\}_{t=1}^{|r_2|}. \tag{7}$$

**Step 3: Similarity score.** The contamination score is the similarity between the two entropy sequences. A higher similarity indicates that the model remains on the same reasoning path despite being instructed to change it, suggesting memorization. We use a length-aware (penalized) cosine similarity:

$$\text{Score}_{\text{Self-Critique}}(q) = \cos_{\text{penalized}}(E_1, E_2), \tag{8}$$

where

$$\cos_{\text{penalized}}(A, B) = \cos\big(\text{pad}(A), \text{pad}(B)\big) \times \frac{\min(|A|, |B|)}{\max(|A|, |B|)}. \tag{9}$$

---

[2]The exact $I_{\text{critique}}$ is shown in Appendix E.

Here $\cos(\cdot, \cdot)$ is the standard cosine similarity between vectors (dot product over the product of L2 norms), and $\text{pad}(\cdot)$ zero-pads the shorter sequence to the maximum length so that non-overlapping positions contribute zero. The multiplicative length ratio penalizes cases where one response is much shorter/longer than the other, since response length itself reflects a facet of the reasoning mode. Overall, a higher score indicates a higher likelihood of contamination. A formal description of the procedure is provided in Appendix A.

## 5 EXPERIMENTS

In this section, we present a comprehensive empirical evaluation of our proposed Self-Critique. We first introduce the RL-MIA benchmark, which we constructed specifically for RL data contamination detection. We then describe the baseline methods we compare against and detail our experimental setup. Finally, we present the main results, followed by analysis and ablation studies. Additional experimental results are in Appendix B and C.

### 5.1 RL-MIA: A BENCHMARK FOR RL MEMBERSHIP INFERENCE ATTACK

To the best of our knowledge, no benchmark currently exists for systematically detecting data contamination during the RL post-training stage. To address this gap, we introduce the RL-MIA (Reinforcement Learning Membership Inference Attack). The key idea behind RL-MIA is to simulate controllable data contamination by selectively including a subset of data in the RL post-training process, while the objective of the contamination detection task is to identify which samples have been used.

The problems used for this simulation are drawn from four benchmarks selected to cover diverse styles and potential pre-training exposure. We include the widely used mathematical reasoning benchmarks, i.e., AIME 2024 and AIME 2025. AIME 2024 may have appeared in some models' pre-training corpora, allowing us to test robustness to prior exposure, whereas AIME 2025 is post-cutoff and thus unlikely to be present in pre-training data. To obtain a controlled setting free from prior exposure, we also include two synthetically generated logical reasoning datasets: Knights & Knaves (K&K) (Xie et al., 2024a) and SAT (Liu et al., 2025). The synthetic nature of these datasets ensures that any detected training signal can be attributed to the RL post-training phase.

To approximate a realistic setting, we embed the selected benchmarks into a larger RL post-training corpus. For AIME24 and AIME25, we use the widely adopted OpenR1-Math-46K (Guha et al., 2025) corpus as the base and inject 50% of each benchmark's items into the RL training data. For K&K and SAT, following their original papers , we use the provided training portions to form the contaminated split and synthesize additional items as held-out clean samples. We primarily use Qwen2.5-7B-Instruct (Qwen et al., 2025) and DeepSeek-Math-7B-Instruct (Shao et al., 2024b) as the model for simulating RL-stage contamination. We also run experiments on Qwen2.5-0.5B-Instruct, Qwen2.5-3B-Instruct (Qwen et al., 2025) and Qwen2.5-7B-Math (Yang et al., 2024). All RL runs are implemented with VeRL (Sheng et al., 2025) framework on $8 \times$ NVIDIA A100 (40 GB). Detailed dataset splits and training settings are provided in Appendix F.

### 5.2 EXPERIMENTAL SETUP

**Baseline Methods** We compare Self-Critique against a set of representative baselines. A key consideration in the RL post-training setting is that training changes the response distribution rather than the likelihood of the prompt. Therefore, for baselines originally designed for pre-training data detection (which operate on input text), we adapt them to operate on the model's *responses* to ensure a fair comparison. The baselines include: ❶ **Perplexity (PPL)** (Gonen et al., 2023), which assumes memorized text has lower perplexity; ❷ **Min-K% Prob** (Shi et al., 2024), which posits that memorized text is less likely to contain low-probability outlier tokens; and ❸ **Min-K%++** (Zhang et al., 2025b), which normalizes token probabilities for a more robust score. We also include ❹ **Recall** (Xie et al., 2024b), which prefixes the text with non-member content and measures the relative change in log-likelihood, and ❺ **CDD** (Dong et al., 2024), which measures output consistency under stochastic sampling via the average token-level edit distance across multiple generations.

We summarize these baselines in Table 1: PPL, Min-K%, and Min-K%++ directly use log-probability properties of the text; Recall and CDD expose differences by injecting a non-member prefix or by randomly sampling multiple outputs, respectively. In contrast, we propose a new probing mechanism, i.e., **self-critique probing**, and use **entropy** as the core signal for RL-stage contamination detection. Following the probing ideas in CDD and Recall, we also introduce two additional entropy-based baselines, ❻ **Entropy-Temp** and ❼ **Entropy-Noise**, which keep the probing mechanisms but replace the consistency/likelihood metric with entropy.

Table 1: A taxonomy of data contamination detection methods. Our work is the first to specifically address the challenges in the RL Post-training phase.

| Method | Probing Mechanism | Core Metric | Designed for |
|---|---|---|---|
| *Existing Methods for Pre-training / SFT* | | | |
| PPL (EMNLP'23) | Intrinsic Property | Log Probability | Pre-training / SFT |
| Min-K% (ICLR'24) | Intrinsic Property | Log Probability | Pre-training / SFT |
| Min-K%++ (ICLR'25) | Intrinsic Property | Log Probability | Pre-training / SFT |
| Recall (EMNLP'24) | Non-member prefix | Log Probability | Pre-training / SFT |
| CDD (ACL'24) | Stochastic Sampling | Edit Distance | Pre-training / SFT |
| *Our Proposed Methods for RL Post-training* | | | |
| Entropy-Temp | Stochastic Sampling | Entropy | RL |
| Entropy-Noise | Non-member prefix | Entropy | RL |
| **Self-Critique(Ours)** | Self-Critique Probing | Entropy | RL |

**Evaluation Metrics** We primarily report the Area Under the ROC Curve (AUC), a standard metric for detection problems such as data contamination and membership inference (Shi et al., 2024; Duan et al., 2024; Zhang et al., 2025b). AUC is threshold-independent and reflects the probability that the detector ranks a randomly chosen contaminated sample higher than a randomly chosen clean one; higher AUC indicates stronger detection performance (50% corresponds to random guess). We also report the F1 score at the Youden threshold (Fluss et al., 2005) as a threshold-specific reference.

## 5.3 MAIN RESULTS

Table 2: Performance of different detection methods on the RL-MIA benchmark across two models. The AVG column is the average AUC across all benchmarks. Best AUC is in **bold**; the second best is underlined.

| Method | AIME24 | | AIME25 | | K&K | | SAT | | AVG |
|---|---|---|---|---|---|---|---|---|---|
| | F1 score | AUC | F1 score | AUC | F1 score | AUC | F1 score | AUC | |
| *Qwen2.5-7B-Instruct* | | | | | | | | | |
| PPL (Gonen et al., 2023) | 0.33 | 0.51 | 0.42 | 0.56 | 0.67 | 0.47 | 0.54 | 0.50 | 0.51 |
| Min-K% (Shi et al., 2024) | 0.59 | 0.49 | 0.52 | 0.44 | 0.70 | 0.54 | 0.32 | 0.50 | 0.49 |
| Min-K%++ (Zhang et al., 2025b) | 0.73 | 0.58 | 0.46 | 0.45 | 0.67 | 0.40 | 0.00 | 0.31 | 0.44 |
| Recall (Xie et al., 2024b) | 0.62 | 0.61 | 0.55 | 0.65 | 0.67 | 0.47 | 0.62 | 0.62 | 0.59 |
| CDD (Dong et al., 2024) | 0.50 | 0.57 | 0.67 | 0.52 | 0.67 | 0.47 | 0.57 | 0.47 | 0.51 |
| Entropy-Temp | 0.73 | 0.64 | 0.12 | 0.42 | 0.59 | 0.49 | 0.66 | 0.69 | 0.56 |
| Entropy-Noise | 0.70 | 0.57 | 0.67 | 0.63 | 0.68 | 0.52 | 0.79 | **0.77** | 0.62 |
| **Self-Critique(Ours)** | 0.69 | **0.72** | 0.76 | **0.72** | 0.69 | **0.66** | 0.69 | 0.67 | **0.70** (↑ **19%**) |
| *DeepSeek-Math-7B-Instruct* | | | | | | | | | |
| PPL (Gonen et al., 2023) | 0.42 | 0.53 | 0.62 | 0.41 | 0.34 | 0.54 | 0.68 | 0.64 | 0.53 |
| Min-K% (Shi et al., 2024) | 0.55 | 0.47 | 0.12 | 0.40 | 0.67 | 0.46 | 0.69 | 0.35 | 0.42 |
| Min-K%++ (Zhang et al., 2025b) | 0.67 | 0.53 | 0.67 | 0.56 | 0.15 | 0.47 | 0.62 | 0.49 | 0.51 |
| Recall (Xie et al., 2024b) | 0.54 | 0.46 | 0.52 | 0.56 | 0.39 | 0.54 | 0.69 | 0.62 | 0.54 |
| CDD (Dong et al., 2024) | 0.30 | 0.49 | 0.65 | 0.51 | 0.08 | 0.48 | 0.66 | 0.50 | 0.50 |
| Entropy-Temp | 0.60 | 0.48 | 0.70 | 0.54 | 0.23 | 0.43 | 0.64 | 0.61 | 0.52 |
| Entropy-Noise | 0.70 | 0.56 | 0.72 | **0.69** | 0.48 | 0.52 | 0.67 | 0.45 | 0.55 |
| **Self-Critique(Ours)** | 0.76 | **0.67** | 0.71 | 0.61 | 0.60 | **0.63** | 0.66 | **0.67** | **0.64** (↑ **19%**) |

The main results of different data contamination methods based on Qwen2.5-7B-Instruct and DeepSeek-Math-7B-Instruct[3] are shown in Table 2. Across both models, **Self-Critique** is the most reliable detector: it attains the best average AUC on Qwen2.5-7B-Instruct (0.70, **+19%** over the best non-ours baseline) and on DeepSeek-Math-7B-Instruct (0.64, also **+19%** improvement), and it leads on most per-dataset AUCs. In contrast, likelihood-based baselines (PPL, Min-K%, Min-K%++) often perform near random guesses and can be unstable. Among existing methods, Recall achieves relatively strong performance (it is also a state-of-the-art approach for pre-training contamination (Xie et al., 2024b)), suggesting that probing to expose membership signals is more effective than relying solely on intrinsic text properties. Moreover, when using the same probing mechanisms, the entropy-based variants perform better (Entropy-Temp vs. CDD; Entropy-Noise vs. Recall), indicating that entropy is a sensitive indicator of RL-induced changes and thus better suited for RL-stage detection.

Finally, compared to random sampling or prefix injection, our self-critique probing aligns more closely with the RL property of dependence on high-reward paths. Within our entropy-based methods, this leads to notable gains, i.e., **+13%** on Qwen2.5-7B-Instruct and **+16%** on DeepSeek-Math-7B-Instruct.

## 5.4 DUAL-STAGE CONTAMINATION IN PRE-TRAINING & RL

To isolate the effects of RL-phase contamination, our previous experiments were primarily conducted on synthetic data or datasets with low levels of contamination. However, for public benchmarks released before an LLM's training cutoff, contamination from both pretraining and the RL phase can co-occur. Therefore, we design a study to distinguish between these two sources of contamination. Concretely, we choose GSM8K (widely acknowledged to suffer from substantial pretraining leakage (Zhang et al., 2024a; Dekoninck et al., 2024; Mirzadeh et al., 2025)) and train Qwen2.5-0.5B-Instruct with the PPO algorithm.[4] We then simulate RL-phase contamination by injecting half of the test set into the RL training data. As discussed in Section 3.2, pretraining (which optimizes via Maximum Likelihood Estimation) and RL (which uses reward-driven policy optimization) pursue different objectives, so their respective forms of contamination are likely to produce distinct effects.

We first assign each test item a pretraining-contamination proxy score using a likelihood-based detector (e.g., PPL) to separate these effects. We then evaluate RL-stage contamination detection under two conditions: ❶ a **lower-pretraining-contamination** subset, created by selecting the bottom-q quantile of items by PPL score (e.g., the lowest 50%); and ❷ a **random-control** subset. To control for any confounding effects from a smaller sample size on AUC, the random-control subset is formed by uniformly sampling the same number of items, thereby matching the subset size while preserving the original data distribution. Our hypothesis predicts that Self-Critique's performance will significantly improve on the former subset, but not on the latter.

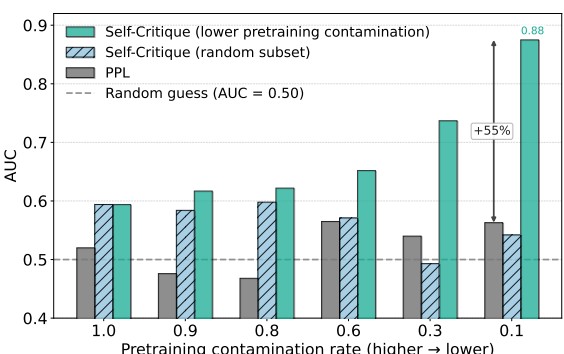

Figure 3: Dual-stage contamination analysis. Self-Critique on the lower-pretraining-contamination subset (green) improves sharply as the rate decreases.

The results are shown in Figure 3, and we also provide the numerical results in Table 7 of Appendix B. As the pretraining contamination level decreases, the performance of Self-Critique on the lower-pretraining-contamination subset improves significantly. In contrast, its performance on the random-control subset shows a slight decrease, while the PPL-based detector's performance approaches that of a random guess. This outcome rules out a pure sample-size effect (as performance on the random-

---

[3]We also provide additional results on Qwen2.5-7B-Math and Llama in Appendix B.

[4]Actually, the setting is the same as the quick start in verl, which makes it easy to reproduce the results.

control subset does not improve) and supports our hypothesis: conditioning the RL detector on items with a weaker pretraining signal allows it to identify RL-phase memorization or path dependence far more clearly. Meanwhile, the fact that likelihood-based cues (PPL) remain unstable and close to 0.5 further demonstrates the effectiveness of our method in specifically targeting RL-phase data contamination.

## 5.5 ABLATION STUDIES

Table 3: Detection performance on K&K with Qwen2.5-3B-Instruct trained by different RL algorithms. AVG column reports the mean AUC across three RL algorithms.

| Method | PPO | | GRPO | | DAPO | | AVG |
|---|---|---|---|---|---|---|---|
| | F1 score | AUC | F1 score | AUC | F1 score | AUC | |
| PPL (Gonen et al., 2023) | 0.65 | 0.41 | 0.63 | 0.46 | 0.61 | 0.51 | 0.46 |
| Min-K% (Shi et al., 2024) | 0.43 | 0.50 | 0.58 | 0.54 | 0.64 | 0.46 | 0.50 |
| Min-K%++ (Zhang et al., 2025b) | 0.39 | 0.50 | 0.61 | 0.52 | 0.26 | 0.49 | 0.51 |
| Recall (Xie et al., 2024b) | 0.61 | 0.46 | 0.35 | 0.49 | 0.55 | 0.50 | 0.48 |
| CDD (Dong et al., 2024) | 0.66 | 0.53 | 0.65 | 0.51 | 0.60 | 0.49 | 0.51 |
| Entropy-Temp | 0.41 | 0.55 | 0.53 | 0.60 | 0.57 | 0.59 | 0.58 |
| Entropy-Noise | 0.64 | 0.59 | 0.43 | 0.52 | 0.60 | 0.49 | 0.53 |
| **Self-Critique(Ours)** | **0.67** | **0.61** | **0.67** | **0.61** | **0.64** | **0.60** | **0.60** (↑ **18%**) |

**Contaminate with Different RL Algorithms**  Table 3 reports results on the K&K task using Qwen2.5-3B-Instruct trained with three RL algorithms, including PPO (Schulman et al., 2017), GRPO (Shao et al., 2024a), and DAPO (Yu et al., 2025). Across all algorithms, **Self-Critique** is the most reliable detector: it attains the best AUC for each algorithm and the highest average AUC (0.60). Entropy-based probes are consistently stronger than likelihood-based methods: Entropy-Temp is the next best on average (0.58), while likelihood baselines (PPL, Min-K%, Min-K%++) are around 0.46–0.51. These trends suggest that our probe, measuring path dependence via entropy similarity, captures an RL-induced signal that is *algorithm-agnostic*. F1 scores follow the same pattern, further indicating that Self-Critique yields both better ranking (AUC) and better thresholded decisions. We also provide a discussion on different alignment algorithms under the RLHF paradigm in Appendix B.3.

**Ablation on Top-$K$ Entropy Approximation**  As LLM vocabularies are large, it is often impractical to obtain the full next-token distribution at every decoding step to compute exact entropy. In many APIs, only Top-$K$ token probabilities are available, so we approximate entropy using those Top-$K$ masses. We conduct an ablation on the choice of $K$ (Table 4). Reducing $K$ does not harm performance; even in the extreme case $K = 3$, the AUC drops only slightly. We attribute this to the long-tailed nature of next-token distributions: most probability mass concentrates on a small set of tokens, and the tail contributes little to entropy. Hence, Top-$K$ entropy is both efficient and accurate enough for our detector. Considering practical applications, we also include a discussion on inference cost in Appendix G.2.

Table 4: Ablation on Top-$K$ entropy approximation (Qwen2.5-7B-Instruct). We report AUC for different $K$ and the row-wise variance across $K \in \{3, 5, 10, 20, 50\}$.

| Dataset | $K=3$ | $K=5$ | $K=10$ | $K=20$ | $K=50$ | Variance |
|---|---|---|---|---|---|---|
| AIME25 | 0.7022 | 0.7111 | 0.7111 | 0.7156 | 0.7156 | $2.39 \times 10^{-5}$ |
| K&K | 0.6460 | 0.6572 | 0.6636 | 0.6608 | 0.6584 | $3.62 \times 10^{-5}$ |

We also provide additional ablation studies about why self-critique probing is better, the sampling strategy and sensitivity to meta-instructions in Appendix C.

## 6 CONCLUSION

In this paper, we presented the first systematic study of data contamination in the overlooked RL post-training stage, demonstrating that existing data contamination detection methods are ill-suited

for this reward-driven paradigm. To address this gap, we proposed Self-Critique, a novel method that identifies RL-induced policy collapse by actively probing the model's reasoning path dependencies using token-level entropy. To validate our method in a controlled setting, we also developed RL-MIA, a new benchmark for RL-phase contamination. Experiments show that Self-Critique consistently outperforms baselines, improving the average AUC by up to 30%, and its ability to isolate RL-specific signals is further highlighted in dual-contamination scenarios, where performance improves by up to 55%. As the community's understanding of RL post-training grows, we expect that more detectors tailored to this unique setting will emerge.

## ACKNOWLEDGEMENT

This research is supported by the National Key R&D Program under Grant No.2023YFB4503801, the National Natural Science Foundation of China under Grant No.62192733, 62192730, 62192731, the Major Program(JD) of Hubei Province (No.2023BAA024), the Beijing Major Science and Technology Project under Contract no.Z251100008425005.

## ETHICS STATEMENT

All authors have read and adhered to the ICLR Code of Ethics. The primary objective of our work is to enhance the integrity and reliability of Large Language Model evaluations. We address this by developing Self-Critique, a method to detect data contamination in the Reinforcement Learning (RL) post-training phase, aiming to prevent misleading performance claims and promote transparent research practices. Our method is a specific application of Membership Inference Attack (MIA) techniques, and we acknowledge the potential dual-use concerns regarding data privacy. However, our study is carefully scoped to mitigate these risks. The goal of our work is defensive—providing a validation tool for researchers—and it is applied exclusively to public, non-sensitive benchmarks (AIME, K&K, SAT, GSM8K) that contain no personal data. We believe the societal benefit of ensuring robust and honest model evaluation significantly outweighs the minimal risk of misuse in this context.

## REPRODUCIBILITY STATEMENT

We are committed to ensuring the full reproducibility of our research. To facilitate this, we have made our code and the newly constructed RL-MIA benchmark available at `https://github.com/yongding-tao/RL-Data-Contamination`. This repository includes the implementation of our proposed Self-Critique method, baseline methods, and scripts to run the experiments. A formal, step-by-step description of the Self-Critique algorithm is provided in Appendix A (Algorithm 1). The specific prompts used for the self-critique probing mechanism are detailed in Appendix E. Details regarding the construction of the RL-MIA benchmark, including dataset sources, injection methods, and data splits, are described in Section 5.1 and further elaborated in Appendix F (Table 6). The key training hyperparameters for all RL models and algorithms are provided in Appendix F (Table 7), ensuring that our training process can be accurately replicated. Our evaluation metrics are standard in the field and are defined in Section 5.1.

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

APPENDIX CONTENTS

# A  ALGORITHM OF SELF-CRITIQUE

---

**Algorithm 1** Self-Critique Contamination Detection

---

**Require:** Model $\mathcal{M}$ (black-box with per-token log-prob access or top-$k$ probs), meta-prompt $I_{\text{critique}}$, problem $q$
**Ensure:** Contamination score $\text{Score}(q)$
    **Deterministic decoding.** Use greedy decoding (temperature $= 0$) in all generations.

1: **(Initial response)** Construct $P_1 \leftarrow T(q)$ and generate $r_1 \leftarrow \mathcal{M}(P_1)$.
2: Compute token-level entropy sequence $E_1 \leftarrow \{H_t(r_1)\}_{t=1}^{|r_1|}$ using per-token probabilities.
3: **(Self-critique response)** Form $q' \leftarrow q \oplus I_{\text{critique}}(r_1)$; construct $P_2 \leftarrow T(q')$ and generate $r_2 \leftarrow \mathcal{M}(P_2)$.
4: Compute entropy sequence $E_2 \leftarrow \{H_t(r_2)\}_{t=1}^{|r_2|}$.
5: **(Similarity)** Pad the shorter sequence with zeros: $\tilde{E}_1 \leftarrow \text{pad}(E_1)$, $\tilde{E}_2 \leftarrow \text{pad}(E_2)$.
6: Compute cosine similarity $s \leftarrow \cos(\tilde{E}_1, \tilde{E}_2)$.
7: Compute length penalty $\lambda \leftarrow \frac{\min(|E_1|, |E_2|)}{\max(|E_1|, |E_2|)}$.
8: **return** $\text{Score}(q) \leftarrow s \times \lambda$.

---

# B  ADDITIONAL RESULTS

## B.1  ADDITIONAL RESULTS ON OTHER MODELS

To better demonstrate the generalization of our method across different models, we provide additional experiments on Qwen2.5-7B-Math (Yang et al., 2024) and Llama-3.1-8B-Instruct (Grattafiori et al., 2024). As shown in Table 5, on Qwen2.5-7B-Math, Self-Critique achieves the best AUC on both AIME24 (0.76) and AIME25 (0.72), and the highest average AUC (0.74). The two entropy baselines are competitive, while likelihood-based methods (PPL, Min-K%, Min-K%++) and Recall/CDD are clearly weaker here. We do not report K&K and SAT for this model because we could not get stable RL training on these synthetic datasets with a non-instruct base; such setups typically require more careful RL hyperparameters and data curation. This engineering detail is outside the scope of our study. The experimental results on Llama-3.1-8B-Instruct are presented in Table 6. Consistent with our previous findings, Self-Critique achieves the highest AUC on both K&K (0.61) and SAT (0.62) datasets, significantly outperforming likelihood-based baselines which remain close to random guessing.

Table 5: Detection results on Qwen2.5-7B-Math under RL-MIA (higher is better). AVG reports mean AUC across AIME24 and AIME25.

| Method | AIME24 | | AIME25 | | AVG |
|---|---|---|---|---|---|
| | F1 score | AUC | F1 score | AUC | |
| PPL(Gonen et al., 2023) | 0.71 | 0.59 | 0.60 | 0.55 | 0.57 |
| Min-K%(Shi et al., 2024) | 0.72 | 0.41 | 0.63 | 0.45 | 0.43 |
| Min-K%++(Zhang et al., 2025b) | 0.69 | 0.48 | 0.58 | 0.46 | 0.47 |
| Recall(Xie et al., 2024b) | 0.45 | 0.55 | 0.32 | 0.49 | 0.52 |
| CDD(Dong et al., 2024) | 0.67 | 0.51 | 0.67 | 0.43 | 0.47 |
| Entropy-Temp | 0.72 | 0.50 | 0.70 | 0.63 | 0.56 |
| Entropy-Noise | 0.70 | 0.64 | 0.67 | 0.51 | 0.57 |
| **Self-Critique (Ours)** | **0.81** | **0.76** | **0.71** | **0.72** | **0.74 (↑ 30%)** |

## B.2  NUMERICAL RESULTS FOR DUAL-STAGE CONTAMINATION

For completeness, we provide the numerical data corresponding to the dual-stage contamination analysis presented in Figure 3 of the main paper. The results, detailed in Table 7, quantify the trend shown in the figure: as the pretraining contamination signal is reduced (i.e., when evaluating on

Table 6: Detection results on Llama-3.1-8B-Instruct under RL-MIA (higher is better). AVG reports mean AUC across K&K and SAT. Self-Critique consistently outperforms baselines on both logic datasets.

| Method | K&K | | SAT | | AVG |
|---|---|---|---|---|---|
| | F1 score | AUC | F1 score | AUC | |
| PPL(Gonen et al., 2023) | 0.52 | 0.47 | 0.67 | 0.53 | 0.50 |
| Min-K%(Shi et al., 2024) | 0.66 | 0.50 | 0.50 | 0.46 | 0.48 |
| Min-K%++(Zhang et al., 2025b) | 0.45 | 0.44 | 0.51 | 0.47 | 0.46 |
| Recall(Xie et al., 2024b) | 0.67 | 0.53 | 0.68 | 0.55 | 0.54 |
| CDD(Dong et al., 2024) | 0.67 | 0.53 | 0.67 | 0.51 | 0.52 |
| Entropy-Temp | 0.68 | 0.56 | 0.67 | 0.53 | 0.55 |
| Entropy-Noise | 0.67 | 0.52 | 0.68 | 0.57 | 0.55 |
| **Self-Critique (Ours)** | **0.69** | **0.61** | **0.70** | **0.62** | **0.62 (↑ 15%)** |

subsets with lower PPL scores), the AUC of Self-Critique on these filtered subsets increases sharply from 0.59 to 0.88, confirming its effectiveness at isolating RL-specific signals.

Table 7: Numerical AUC results for the dual-stage contamination analysis. The header row indicates the quantile of pretraining contamination retained.

| Method | 1.0 | 0.9 | 0.8 | 0.6 | 0.3 | 0.1 |
|---|---|---|---|---|---|---|
| PPL | 0.52 | 0.48 | 0.47 | 0.56 | 0.54 | 0.56 |
| Self-Critique (random subset) | 0.59 | 0.58 | 0.60 | 0.57 | 0.49 | 0.54 |
| **Self-Critique (lower pretraining contamination)** | **0.59** | **0.62** | **0.62** | **0.65** | **0.74** | **0.88** |

### B.3 ADDITIONAL RESULTS ON RLHF PARADIGM

While our primary investigation in the main text focuses on the Reinforcement Learning with Verifiable Rewards (RLVR) paradigm due to its rising prominence in reasoning tasks, we acknowledge the critical role of Reinforcement Learning with Human Feedback (RLHF) in the broader post-training landscape. To demonstrate the generalizability of our method, we extended our experiments to the standard RLHF setting. Following the experimental setup of Zhong et al. (2025), we utilized Llama-3-8B-Instruct trained on the UltraFeedback dataset (Cui et al., 2023) under four distinct alignment algorithms: PPO (Schulman et al., 2017) (representing traditional reward modeling), DPO (Rafailov et al., 2023) (representing implicit reward alignment), TDPO (Zeng et al., 2024b) and RTO (Zhong et al., 2025) (representing token-level reward modeling).

Table 8: Detection results (AUC) on the UltraFeedback dataset across different RLHF alignment algorithms. Self-Critique consistently outperforms baselines regardless of the alignment method.

| Method | PPO (Schulman et al., 2017) | DPO (Rafailov et al., 2023) | TDPO (Zeng et al., 2024b) | RTO (Zhong et al., 2025) |
|---|---|---|---|---|
| PPL(Gonen et al., 2023) | 0.56 | 0.45 | 0.49 | 0.53 |
| Min-K%(Shi et al., 2024) | 0.40 | 0.53 | 0.49 | 0.45 |
| Min-K%++(Zhang et al., 2025b) | 0.47 | 0.52 | 0.51 | 0.44 |
| Recall(Xie et al., 2024b) | 0.58 | 0.58 | 0.60 | 0.56 |
| CDD(Dong et al., 2024) | 0.56 | 0.46 | 0.44 | 0.44 |
| Entropy-Temp | 0.54 | 0.53 | 0.54 | 0.52 |
| Entropy-Noise | 0.53 | 0.51 | 0.53 | 0.59 |
| **Self-Critique (Ours)** | **0.62** | **0.62** | **0.64** | **0.70** |

The detection results are presented in Table 8. We observe that Self-Critique consistently maintains its effectiveness across all four paradigms, achieving the highest AUC scores ranging from 0.62 to 0.70. In contrast, baseline methods such as PPL and CDD often perform near random guessing (AUC ≈ 0.50) or exhibit instability. This suggests that the phenomenon of policy collapse is a

fundamental characteristic of alignment training, regardless of whether the reward signal is sparse, implicit, or token-level.

## C  ADITIONAL ABLATIONS

### C.1  WHY SELF-CRITIQUE PROBING IS A BETTER DETECTOR

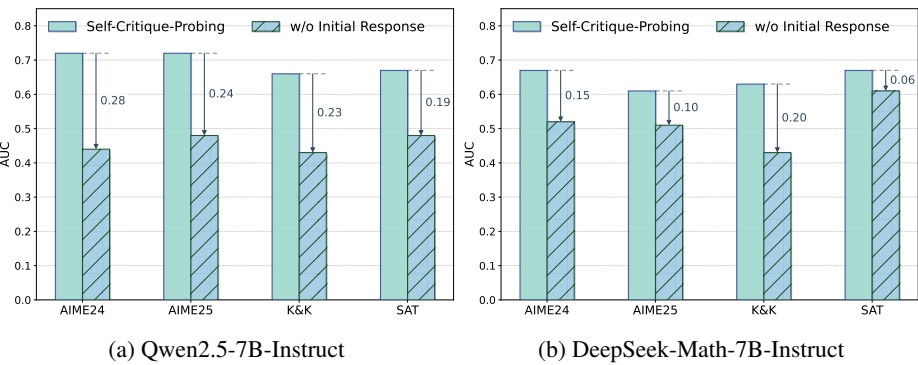

(a) Qwen2.5-7B-Instruct             (b) DeepSeek-Math-7B-Instruct

Figure 4: Self-critique probing vs. no self-critique.

As our self-critique idea asks the model to propose an alternative reasoning path given its initial response, a naive variant is to skip the initial response and simply tell the model to "answer using an unusual technique"[5]. The results in Figure 4 show that without the initial response as an anchor, performance collapses to near random guess . This variant fails because it removes the anchor that makes the probe meaningful: without conditioning on the initial answer, the "alternative" path is unconstrained, instruction following is noisy, and both members and non-members produce heterogeneous trajectories whose entropy sequences are not comparable. Moreover, RL can induce mode-seeking even on unseen items, so we end up measuring two arbitrary paths rather than the deviation from a memorized one. Self-critique fixes a concrete baseline and probes deviation from it—hence the large AUC gains over the no-response variant.

### C.2  ABLATION ON SAMPLING STRATEGY

We perform an ablation study of the sampling strategies used to generate model responses. Specifically, we experiment with three settings: (1) using greedy sampling for both the initial response and the second self-critique response; (2) using greedy sampling for the initial response and temperature sampling for the self-critique response; and (3) using temperature sampling for both the initial response and the self-critique response. For temperature sampling, we test multiple temperatures. The ablation results, shown in Figure 5, indicate that the best performance is achieved when both the initial and self-critique responses are generated by greedy sampling. This is because greedy sampling eliminates the effect of randomness, thereby better revealing the sharp policy distributions caused by entropy collapse from RL post-training.

### C.3  ABLATION ON META-INSTRUCTION

To address the concern that our method's behavior might be tied to a specific prompt template, we conducted an ablation study to test its sensitivity to the Self-Critique instruction. We evaluated five paraphrased variations of our original meta-instruction (detailed as Variations 1–5 in Appendix E).

The results, presented in Table 9, demonstrate that Self-Critique is exceptionally stable across different templates. The standard deviation in AUC across all meta-prompts is remarkably low (0.0251 on AIME25 and 0.0254 on K&K). This quantitative evidence confirms that the detection signal is driven by the underlying policy collapse mechanism rather than the specific phrasing of the instruction.

---

[5]Detail prompt is shown in Appendix E Prompt 1

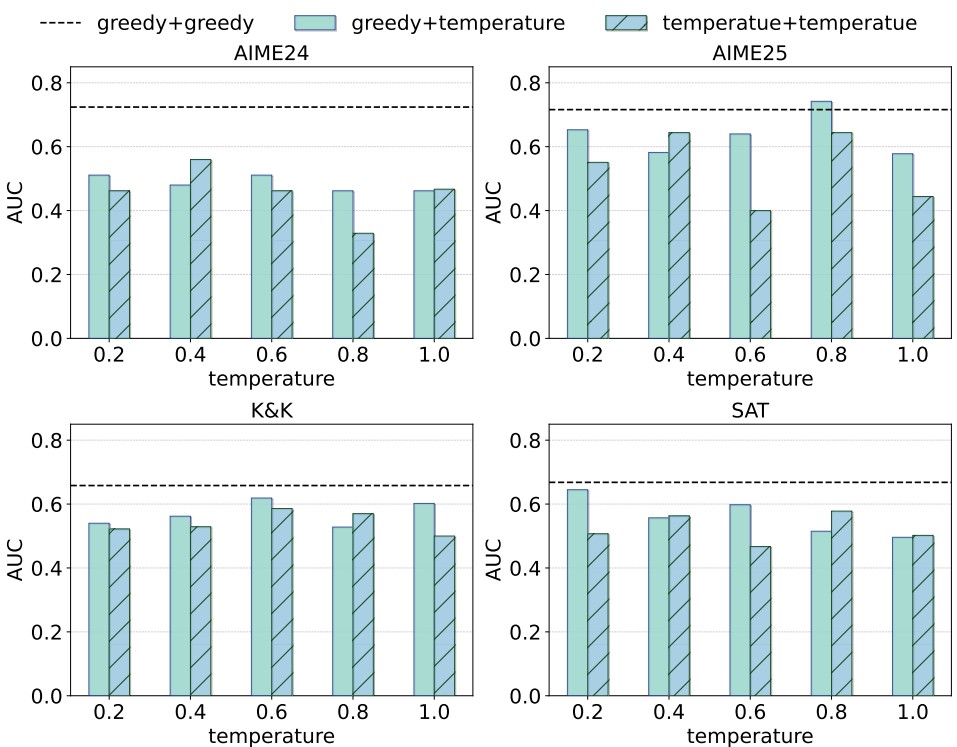

Figure 5: Ablation on Sampling Strategy

Table 9: Ablation study on different Self-Critique meta-instructions. The method shows high robustness to prompt variations, as indicated by the low standard deviation.

| Meta-Instruction | AIME25 | | K&K | |
|---|---|---|---|---|
| | F1 score | AUC | F1 score | AUC |
| Original | 0.7647 | 0.7156 | 0.6866 | 0.6584 |
| Variation 1 | 0.7179 | 0.6978 | 0.6992 | 0.6464 |
| Variation 2 | 0.7879 | 0.7600 | 0.6763 | 0.6432 |
| Variation 3 | 0.7429 | 0.6978 | 0.7302 | 0.7016 |
| Variation 4 | 0.7500 | 0.7422 | 0.7009 | 0.6640 |
| Variation 5 | 0.7500 | 0.7333 | 0.6809 | 0.6272 |
| **Mean** | 0.7522 | **0.7244** | 0.6957 | **0.6568** |
| **Std** | 0.0233 | **0.0251** | 0.0195 | **0.0254** |

## D    VISUALIZATION OF CONTAMINATION SCORE DISTRIBUTION

For the contaminated and uncontaminated samples in the AIME and AIME25 dataset, we computed their Self-Critique similarity scores, and present the histograms in Figure 6. Through Kernel Density Estimation (KDE), we observe a clear difference in the distribution of Self-Critique similarity scores between contaminated and uncontaminated samples, demonstrating the effectiveness of our proposed Self-Critique method for data contamination detection.

## E    INSTRUCTIONS

We present here the prompts used in the data contamination detection process. We employ Prompt 1 to encourage the model to generate responses that deviate from conventional reasoning. In particular,

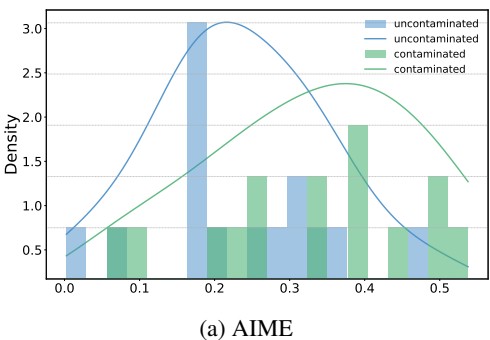 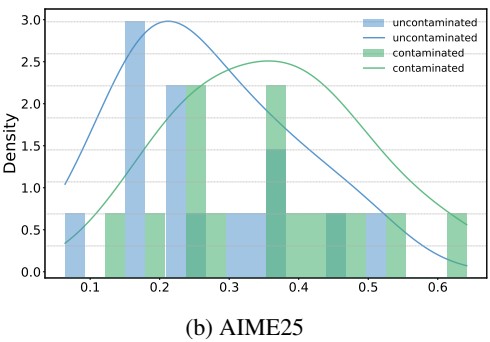

(a) AIME             (b) AIME25

Figure 6: Histograms and Kernel Density Estimation (KDE) of Self-Critique similarity scores between contaminated and uncontaminated samples.

for the Self-Critique process, we use Prompt 2 to guide the model to regenerate a second response that differs as much as possible from the initial one. For the ablation study on template sensitivity, we additionally employed five paraphrased variations (Variations 1–5) detailed below.

---

**Prompt 1: Unconventional Reasoning**

Answer using a technique you'd typically avoid or a deliberately unconventional line of reasoning.

---

**Prompt 2: Self-Critique Instruction (Original)**

A possible answer is provided below (it may or may not be correct). Please provide a response that follows a different reasoning path or provides an alternative solution:
—
{Initial Response}
—
Please now provide your new, different response:

---

**Prompt 3: Self-Critique Variation 1**

A potential answer is shown below (it may or may not be correct). Please give a response that uses a different reasoning approach or offers an alternative solution:
—
{Initial Response}
—
Please now give your new, different response:

---

**Prompt 4: Self-Critique Variation 2**

An answer attempt is provided below (it may or may not be correct). Please provide a response that takes a different reasoning path or presents an alternative solution:
—
{Initial Response}
—
Please now provide your new, alternative response:

> **Prompt 5: Self-Critique Variation 3**
>
> A candidate answer is provided below (it may or may not be correct). Please give a response that employs a different reasoning path or supplies an alternative solution:
> —
> {Initial Response}
> —
> Please now give your new, distinct response:

> **Prompt 6: Self-Critique Variation 4**
>
> A potential solution is presented below (it may or may not be correct). Please provide a response that adopts a different reasoning path or offers an alternative solution:
> —
> {Initial Response}
> —
> Please now provide your new, varied response:

> **Prompt 7: Self-Critique Variation 5**
>
> A tentative answer is provided below (it may or may not be correct). Please provide a response that uses a different reasoning path or presents an alternative solution:
> —
> {Initial Response}
> —
> Please now provide your new, alternative response:

# F   BENCHMARK DETAIL & TRAINING SETTINGS

This section provides a detailed breakdown of the RL-MIA benchmark construction and the specific training configurations used in our experiments.

To create a controlled environment for evaluating data contamination in the RL phase, we constructed the RL-MIA benchmark. The methodology involves injecting a known subset of test samples (contaminated items) into a larger base corpus for RL post-training. The detection task is then to distinguish these injected samples from clean, unseen samples. Table 10 summarizes the data splits for each source dataset, detailing the size of the base RL corpus, the number of injected items, the total size of the final training set, the number of times each contaminated item appears (Occurrences), and the total number of items in the final detection task (Contaminated + Clean).

For reproducibility, we also provide the key hyperparameters used during RL post-training. Table 11 lists the shared training parameters for our two primary experimental models: Qwen2.5-7B-Instruct and Deepseek-math-7b-Instruct.

Table 10: RL-MIA data splits for training and evaluation. The last column reports the total number of evaluation items (Contam + Clean).

| Source | Base RL Corpus (size) | Injected items | Train Size | Occurrences | Detection Tasks |
|--------|----------------------|----------------|------------|-------------|-----------------|
| **AIME24** | OpenR1-Math-46K | **15** | 46K + 30 | **2** | **30** |
| **AIME25** |  | **15** |  |  | **30** |
| **K&K** | K&K train: **950** | K&K test: **50** | 950 + 50 | **3** | **100** |
| **SAT** | SAT train: **450** | SAT test: **50** | 450 + 50 |  | **100** |
| **GSM8K** | GSM8K train: **7473** | GSM8K test: **659** | 7473 + 659 | **4** | **1319** |

Due to the 4,096 context length of DeepSeek-Math-7B-Instruct, we set its maximum generation length to 3,072 (with a 1,024-token prompt). For other models (Qwen2.5-7B-Math, Qwen2.5-

Table 11: Shared training hyperparameters

| Parameter | Qwen2.5-7B-Instruct | Deepseek-Math-7B-Instruct |
|---|---|---|
| Actor learning rate | $1.0 \times 10^{-6}$ | $1.0 \times 10^{-6}$ |
| Batch size (train / val) | 128 / 512 | 128 / 512 |
| Max prompt length | 1024 | 1024 |
| Max generation length | 4096 | 3072 |
| Temperature (train / val) | 1.0 / 0.6 | 1.0 / 0.6 |
| Samples per prompt ($n$) | 8 | 8 |
| Tensor model parallel (TP) | 2 | 2 |
| micro / mini-batch | 2 / 2 | 2 / 2 |
| Max tokens per GPU | 16384 | 16384 |
| Use KL loss | No | No |
| Entropy coefficient | 0.001 | 0.001 |

3B-Instruct, and Qwen2.5-0.5B-Instruct), the settings are essentially the same as for Qwen2.5-7B-Instruct. Full details are available in the training scripts included with our released code.

# G  DISCUSSION

## G.1  DISCUSSION ON RESULT VARIABILITY AND ERROR BARS

In Table 2, we observed variability in the performance of baseline methods across different models, particularly for Entropy-Noise on the SAT dataset. This variability largely stems from the nature of the probing mechanism: Entropy-Noise injects a random, non-member prefix to disrupt the context. Different models react differently to these out-of-distribution prefixes—some models robustly ignore them, while others may become unstable or hallucinate, leading to erratic entropy shifts that do not consistently correlate with contamination. In contrast, Self-Critique uses a semantically meaningful instruction, guiding the model into a more predictable state, which results in more stable detection.

Table 12: Bootstrap analysis (1,000 resamples) on the SAT dataset. We report Mean ± Std and the [95% Confidence Interval]. Self-Critique shows stable performance significantly above random guessing.

| Method | Qwen2.5-7B-Instruct | | DeepSeek-Math-7B | |
|---|---|---|---|---|
| | Mean ± Std | 95% CI | Mean ± Std | 95% CI |
| PPL(Gonen et al., 2023) | 0.50 ± 0.06 | [0.38, 0.63] | 0.64 ± 0.06 | [0.52, 0.75] |
| Min-K%(Shi et al., 2024) | 0.50 ± 0.06 | [0.37, 0.61] | 0.35 ± 0.06 | [0.24, 0.47] |
| Min-K%++(Zhang et al., 2025b) | 0.31 ± 0.05 | [0.21, 0.42] | 0.49 ± 0.06 | [0.38, 0.61] |
| Recall(Xie et al., 2024b) | 0.62 ± 0.06 | [0.51, 0.72] | 0.62 ± 0.06 | [0.50, 0.74] |
| CDD(Dong et al., 2024) | 0.47 ± 0.06 | [0.36, 0.58] | 0.50 ± 0.02 | [0.46, 0.54] |
| Entropy-Temp | 0.69 ± 0.05 | [0.58, 0.78] | 0.61 ± 0.06 | [0.49, 0.72] |
| Entropy-Noise | 0.77 ± 0.05 | [0.67, 0.86] | 0.45 ± 0.06 | [0.33, 0.56] |
| **Self-Critique (Ours)** | **0.67 ± 0.05** | **[0.56, 0.76]** | **0.67 ± 0.06** | **[0.55, 0.77]** |

To rigorously quantify this stability, we performed a bootstrap analysis (Efron, 1992) (resampling 1,000 times) to calculate the mean AUC and 95% Confidence Intervals (CI) (DiCiccio & Efron, 1996) for the SAT dataset. As shown in Table 12, while baseline performance fluctuates significantly (e.g., Min-K%++ varies from 0.21 to 0.42 on Qwen), Self-Critique demonstrates robust performance with confidence intervals that consistently surpass the likelihood-based methods.

## G.2  DISCUSSION ON INFERENCE COST

A practical consideration for deployment is the trade-off between detection performance and computational budget. We categorize detection methods into *passive* (e.g., PPL, Min-K%) and *active* (e.g., Recall, CDD, Self-Critique) approaches. ❶ **Passive Methods:** These are computationally cheapest, requiring only a single forward pass of the text. However, as demonstrated in our experiments (Table

2), they are largely ineffective for RL post-training contamination because the likelihood signal is decoupled from the reward-driven training objective. ❷ **Active Methods:** These require additional computation to probe the model but are necessary for valid detection in this regime. Among active methods, Self-Critique offers the best balance. It requires two generations (initial + critique). In comparison, Recall also requires two passes but generally achieves lower AUC. CDD typically requires a large number of samples (e.g., $N = 50$) to estimate the edit distance distribution reliably. If we limit CDD to just two samples to match Self-Critique's budget, its estimation of the edit distance distribution becomes noisy and unreliable, leading to meaningless results.

Regarding the overhead of computing token-level entropy, our ablation study in Table 4 demonstrates that approximating entropy using only the Top-$K$ (e.g., $K = 5$) probabilities is sufficient. This avoids the need to materialize the full vocabulary distribution, making the entropy calculation overhead negligible compared to the generation cost.

## H    LIMITATIONS AND FUTURE WORK

While our work presents a robust framework for detecting contamination in the RL post-training phase, we acknowledge several avenues for future research that build upon our findings.

**Generalization Across Diverse Domains.**    Our experiments primarily focused on mathematical and logical reasoning tasks, as these are prominent domains where RL has demonstrated significant benefits. However, the unique characteristics of RL-induced contamination may vary across different problem domains. For instance, in areas such as code generation, where a wider diversity of valid solutions is common, the signature of policy collapse might manifest differently. Future work could extend the evaluation of Self-Critique and other reward-aware detection methods to these and other domains, thereby assessing the broader applicability and potential domain-specific adaptations of our method.

**Scalability to Larger Models.**    The models used in our study, ranging from 0.5B to 7B parameters, are representative of a widely used class of open-source LLMs. However, the landscape of foundation models is rapidly evolving, with state-of-the-art proprietary and open-source models now exceeding hundreds of billions of parameters. While we have no reason to believe our method's core principles would not apply, the dynamics of policy collapse and memorization at such scales are not yet fully understood. Investigating the effectiveness of Self-Critique on these frontier models represents an important next step in ensuring the reliability of the entire LLM ecosystem.

## I    LLM USAGE STATEMENT

In preparing this manuscript, we use LLMs to aid and polish the writing. Specifically, LLMs improve clarity, grammar, and phrasing, ensuring the text is concise and readable. The use of LLMs does not influence the technical contributions or the interpretation of experimental findings. All content polished by LLMs is carefully checked by the authors.

