# OpenReview forum: "Detecting Data Contamination from Reinforcement Learning Post-training for Large Language Models"
_ICLR.cc/2026/Conference — ICLR 2026 Poster_

### Official Review · Reviewer_PKgz · 2025-10-24

**Soundness:** 3
**Presentation:** 3
**Contribution:** 3
**Rating:** 6
**Confidence:** 2

**Summary:**

This paper introduces a membership inference attack strategy for models trained on  reinforcement learning with feedback via the exploitation of the entropy collapse in RL learned policies. The work relies on the observation that the RL learned response traces often suffer from entropy collapse on high reward regions. The key idea behind the work is to use the length penalized entropy measure between an original prompt’s response and the response generated with an added critique meta prompt as a mechanism for member detection. The paper showcases that in RL trained policies the proposed method performs better than the existing works which were designed towards next token prediction training objectives (SFT, Pretraining etc)

**Strengths:**

1. The paper is easy to follow and is well written.

2. Dataset and model diversity is present in the experimentation.

3. To my knowledge this is the first MIA work reinforcement learning alignment. But my expertise in MIA attacks is also limited.

4. The proposed method showcases effective performance as opposed to other baselines which are designed towards models trained with maximum likelihood.

5. A certain level of ablation has been performed on different RL methods under Section 5.4.

**Weaknesses:**

1. The method seems to be motivated by the sparse reward assumption of verifiable reward. Human feedback based preference datasets still play a vital role in capturing human feedback. While traditional reward models are learned via the Bradley Terry formulation on a complete response there, token level reward modelling has also gained prominence. Further implicit reward based alignment such as DPO also exists. Can you provide the prospects of the generalizability of the method across these different alignments.

2. Can you please provide further details about the reward models used for the RL training under the section “Contaminate in Different RL algorithms”

**Questions:**

See weaknesses

---

> ### Author Response · Authors · 2025-11-22
> **Author Response**
>
> We sincerely thank you for your thoughtful feedback and for recognizing the importance of our work in addressing RL post-training contamination. We will address the noted weaknesses and questions below. **We also provide a revised pdf with new content marked in blue.**
>
> `W1: The method seems to be motivated by the sparse reward assumption of verifiable reward. Human feedback based preference datasets still play a vital role in capturing human feedback... Can you provide the prospects of the generalizability of the method across these different alignments.`
>
> We appreciate you raising this crucial point regarding the scope of alignment techniques. While our initial focus was on Reinforcement Learning with Verifiable Rewards (RLVR) due to its rising prominence in reasoning tasks, we fully agree that Reinforcement Learning with Human Feedback (RLHF) is equally important in the post-training landscape.
>
> To address this, we extended our experiments to include the standard RLHF setting. Following the experimental setup of the **RTO** paper [4], we utilized the  Llama-3-8B-Instruct trained on UltraFeedback dataset under four different alignment algorithms: **PPO** [1] (representing traditional reward modeling), **DPO** [2] (representing implicit reward alignment),  **TDPO** [3] and **RTO** [4] (representing token-level reward modeling).
>
> The detection results (AUC) are presented in the table below. The results show that Self-Critique maintains its effectiveness across all four paradigms. While baseline methods like PPL and CDD often hover near random guessing (AUC $\approx$ 0.50) or show instability, Self-Critique consistently achieves the highest AUC scores, ranging from 0.62 on PPO/DPO to 0.70 on RTO. This suggests that the phenomenon of policy collapse is a fundamental characteristic of alignment training, whether the reward signal is sparse, implicit, or token-level. We have included these results in **Appendix B.3** of the revised PDF.
>
> | Method | PPO | DPO | TDPO | RTO |
> | :--- | :---: | :---: | :---: | :---: |
> | PPL | 0.56 | 0.45 | 0.49 | 0.53 |
> | Min-K% | 0.40 | 0.53 | 0.49 | 0.45 |
> | Min-K%++ | 0.47 | 0.52 | 0.51 | 0.44 |
> | Recall | 0.58 | 0.58 | 0.60 | 0.56 |
> | CDD | 0.56 | 0.46 | 0.44 | 0.44 |
> | Entropy-Temp | 0.54 | 0.53 | 0.54 | 0.52 |
> | Entropy-Noise | 0.53 | 0.51 | 0.53 | 0.59 |
> | **Self-Critique (Ours)** | **0.62** | **0.62** | **0.64** | **0.70** |
>
> `W2: Can you please provide further details about the reward models used for the RL training under the section “Contaminate in Different RL algorithms”`
>
> For the experiments in the "Contaminate with Different RL Algorithms" paragraph of section 5.5 (on the K&K dataset), we utilized a rule-based verifiable reward system rather than a learned neural reward model. This approach is standard for mathematical and logical reasoning tasks where ground truth is deterministic. The reward function acts as a program-based judge that evaluates the model's output in three distinct steps.
>
> First, the function performs a structural validation to ensure the model follows the required format, specifically checking for the presence and correct ordering of \<think\> and \<answer\> tags. Failure to adhere to this structure results in a penalty. Second, it parses the content within the answer tags to extract the predicted roles (e.g., identifying which character is a Knight or a Knave). Finally, it compares these extracted predictions against the ground truth solution dictionary. The model receives a high positive reward (+2.0) only if the answer is a full match with the ground truth; otherwise, it receives a negative reward (-1.5 for incorrect answers or -2.0 for parsing failures). This ensures the RL process is strictly optimizing for logical correctness and instruction following.
>
> References:
>
> [1] Schulman et al. Proximal Policy Optimization Algorithms. arXiv:1707.06347
>
> [2] Rafailov et al. Direct Preference Optimization: Your Language Model is Secretly a Reward Model. NeurIPS 2023
>
> [3] Zeng et al. Token-level Direct Preference Optimization. ICML2024
>
> [4] Zhong et al. DPO Meets PPO: Reinforced Token Optimization for RLHF. ICML2025
>
> -----
>
> Thanks again for your time and for these important points，your feedback has been a great help in strengthening our paper.

---

> > ### Comment · Reviewer_PKgz · 2025-11-23
> > **Response to the authors**
> >
> > I thank the authors for their response and additional results on other RLHF paradigm. My concerns have been addressed. I am maintaining my rating leaning towards accept

---

> > > ### Author Response · Authors · 2025-11-26
> > >
> > > Thank you for your response, and we are glad that we have addressed the concerns. If there are any further questions, we are also happy to discuss.

---

### Official Review · Reviewer_5NYt · 2025-11-01

**Soundness:** 2
**Presentation:** 3
**Contribution:** 3
**Rating:** 4
**Confidence:** 4

**Summary:**

This paper addresses the overlooked issue of data contamination during the RL post-training phase of large language models (LLMs). This work explores contamination specific to RL post-training. The authors introduce Self-Critique, an entropy-based probing method that measures the similarity between two model outputs: an initial answer and a self-critique re-generation conditioned on that answer. A high similarity in entropy curves between the two is interpreted as evidence of contamination, under the hypothesis that contaminated examples lead to policy collapse toward memorized reasoning paths.

**Strengths:**

1. Brings attention to the underexplored problem of contamination during RL post-training, which is an important practical concern.
2. Proposes a simple, reproducible probing method that builds on self-consistency and entropy measurement.
3. Introduces RL-MIA, a useful benchmark that may benefit future work in this area.

**Weaknesses:**

1. Weak Novelty: The proposed “Self-Critique” method is a straightforward application of self-consistency probing combined with entropy comparison. It does not provide a fundamentally new theoretical or algorithmic contribution.
2. Questionable Signal Reliability: The paper assumes that contamination causes entropy path similarity, but similar effects can arise from deterministic reasoning or narrow reward optimization. The entropy path is not guaranteed to distinguish contamination from general RL collapse. And it is doubtable that the method’s behavior is closely tied to the chosen prompt and “self-critique” instruction. In more diverse or flexible RL training settings, the entropy path could vary widely, undermining the stability of the metric.

**Questions:**

1. Why entropy path is a reliable signal and is it sensitive to the template diversity?
2. What is the key difference of the self-critique probing and direct probing with different templates?

---

> ### Author Response · Authors · 2025-11-22
> **Author Response (Part 1/2)**
>
> We sincerely thank you for your thoughtful feedback and for recognizing the importance of our work in addressing RL post-training contamination. We will address the noted weaknesses and questions below. **We also provide a revised pdf with new content marked in blue.**
>
> `W1： Weak Novelty: The proposed “Self-Critique” method is a straightforward application of self-consistency probing combined with entropy comparison. It does not provide a fundamentally new theoretical or algorithmic contribution.`
>
> We appreciate the comment on novelty and would like to clarify the concrete methodological distinctions between our Self-Critique method and the Self-Consistency technique[[1]](https://arxiv.org/pdf/2203.11171). The two methods differ fundamentally in their generation process. Self-Consistency samples a set of **independent**, parallel reasoning paths from a single prompt, extracting the final answer from each path to perform a majority vote. While Our method employs a sequential, **conditional** process: we first generate an initial path, and then generate a second path that is explicitly conditioned on the first.
>
> And we must state that our technical approach, which combines this active probing mechanism with an entropy-based signal, is **the first of its kind specifically to the unique challenges of RL post-training**. As our summary in **Table 1** shows, this establishes a new category of detection methods that moves beyond the likelihood-based signals of prior work.
>
> As you also kindly noted, this is an "important practical concern." We believe **our main novelty lies in identifying this problem and proposing a simple, effective, and reproducible method as an initial solution**(especially compared to previous methods that were close to a random guess). We believe our work, including the introduction of the RL-MIA benchmark, serves as a foundation that will inspire future research into what could be more complex theoretical or algorithmic innovations.
>
> [1] Wang, Xuezhi, et al. "Self-Consistency Improves Chain of Thought Reasoning in Language Models." The Eleventh International Conference on Learning Representations, 2023.
>
> `W2 & Q1 part1： Questionable Signal Reliability: The paper assumes that contamination causes entropy path similarity, but similar effects can arise from deterministic reasoning or narrow reward optimization. The entropy path is not guaranteed to distinguish contamination from general RL collapse.
> Why entropy path is a reliable signal ...?`
>
> Thank you for raising this critical question. Since MIA in the RL phase has not been studied before, we think that an effective detection metric must show a significant change before and after RL training. As you also mentioned, RL training leads to policy collapse, and this behavior is effectively captured by the entropy path. However, using this metric directly was not effective since this collapse exists even in clean samples. We then tried to adopt active probing methods from pre-training contamination detection, like Recall and CDD, by changing their metrics to the entropy path similarity. Although this improved performance, the results were not stable. This led us to design the Self-Critique method. As shown in **Figure 1(a) and 1(b)** (where the x-axis is the position and the y-axis is token-level entropy), contaminated samples show highly similar entropy paths for the initial and critique responses (Figure 1(a)). In contrast, clean samples show different reasoning patterns(Figure 1(b)). We also visualize the distribution differences in **Figure 6 of Appendix D**, which plots the similarity of entropy paths under Self-Critique probing. The clear difference between the contaminated samples (higher scores) and clean samples (lower scores) enables effective detection. Furthermore, our experiments (**Tables 2–6**) consistently show that Self-Critique is the best, while our proposed active probing baselines that use entropy are the second best. This verifies that using entropy path as detection metric is better than log-probability in MIA for RL post-training stage.

---

> ### Author Response · Authors · 2025-11-22
> **Author Response (Part 2/2)**
>
> `W2 & Q1 part 2: it is doubtable that the method’s behavior is closely tied to the chosen prompt and “self-critique” instruction.
> ... is it sensitive to the template diversity?`
>
> Regarding the concern that our method's behavior is tied to a specific prompt, we conducted a new ablation study to test this directly. We evaluated five paraphrased variations of our Self-Critique meta-instruction and found the performance to be exceptionally stable. The standard deviation in AUC across all meta-prompts was **0.0251 on AIME25** and **0.0254 on K&K**. This low std provides quantitative evidence that our method is robust.
>
> Thanks again for pointing out this critical missing point. We have added this ablation study to the **Appendix C.3** in the revised version to further strengthen our claim.
>
> | Meta-Instruction-id | AIME25 F1 score | AIME25 AUC | K&K F1 score | K&K AUC |
> | ------------------- | --------------- | ---------- | ------------ | ------- |
> | Original            | 0.7647          | 0.7156     | 0.6866       | 0.6584  |
> | 1                   | 0.7179          | 0.6978     | 0.6992       | 0.6464  |
> | 2                   | 0.7879          | 0.7600     | 0.6763       | 0.6432  |
> | 3                   | 0.7429          | 0.6978     | 0.7302       | 0.7016  |
> | 4                   | 0.7500          | 0.7422     | 0.7009       | 0.6640  |
> | 5                   | 0.7500          | 0.7333     | 0.6809       | 0.6272  |
> | **mean**            | 0.7522          | **0.7244** | 0.6957       | **0.6568**|
> | **std**             | 0.0233          | **0.0251** | 0.0195       | **0.0254**|
>
> `W2 part 3: In more diverse or flexible RL training settings, the entropy path could vary widely, undermining the stability of the metric.`
>
> Our method remains stable across different RL training dynamics, as shown in **Table 3**. We evaluated Self-Critique on models trained with three distinct RL algorithms: PPO, GRPO, and DAPO. The results show that our method consistently and significantly outperforms all baselines for all three algorithms. This demonstrates that our metric captures a fundamental, algorithm-agnostic signal of contamination.  and is stable across different RL training dynamics.
> Additionally, in response to Reviewer PKgz, we also validate our method on Llama-3 trained with the standard RLHF paradigm, where Self-Critique similarly outperformed baselines, including PPO, DPO, TDPO and RPO. This result is in **Appendix B.3**
>
>
> `Q2: What is the key difference of the self-critique probing and direct probing with different templates?`
>
> The key difference lies in our probing mechanism. Self-Critique probing first generates an initial response, which serves as an anchor. The second response is then explicitly conditioned on this anchor, with the instruction to deviate from it. This creates a constrained and focused test, directly measuring the model's flexibility and dependence on the initial path.
>
> In contrast, direct probing with two different templates would generate two independent, unconstrained reasoning paths. Comparing them would be noisy and less effective for detecting the kind of policy rigidity caused by memorization, as both paths could simply be arbitrary valid solutions. The critical importance of our anchoring mechanism is empirically validated in our ablation study in **Appendix C.1 and Figure 4**, where removing the anchor causes detection performance to drop, confirming our approach is fundamentally different and more effective.
>
>
> -----
>
> Thanks again for your time and for these important points，your feedback has been a great help in strengthening our paper.

---

> > ### Comment · Reviewer_5NYt · 2025-11-28
> >
> > Thank you for the detailed response and additional experiments. My remaining concern is now mainly high-level, rather than empirical:
> >
> > 1. **Necessity and scope of RL-stage contamination detection.**
> >    In realistic RL pipelines, training often uses different prompt templates and similar task families. Where exactly do you draw the line between “legitimate RL on a similar distribution” and “unacceptable contamination” of a specific benchmark instance? RL will generate different rollouts. Is it necessary to detect the RL-stage contamination?
> >
> > 2. **Feasibility under diverse prompt templates.**
> >    If RL training uses diverse meta-instructions / formats that differ from evaluation, is per-instance membership detection still a meaningful and attainable goal, or is your method mainly targeting “strong” contamination where RL prompts are close to the evaluation format? It would be useful to explicitly discuss the limitations of Self-Critique when RL prompt templates are highly heterogeneous.
> >
> > I don’t think more experiments are needed; a short high-level clarification of these points would be enough. If this conceptual scope is made more explicit, I would be inclined to increase my score.

---

> > > ### Author Response · Authors · 2025-11-29
> > >
> > > We sincerely thank you for these insightful high-level questions. They help clarify the boundaries of our work. We discuss each point below:
> > >
> > > **1. Necessity and Scope**:
> > > We agree that RL training on similar distributions is reasonable for data augmentation and improving model capabilities. Therefore, we strictly define "unacceptable contamination" as including **specific benchmark instances** in training, which makes evaluation inaccurate and poses privacy risks.
> > >
> > > Our experiments on synthetic logic datasets (K&K and SAT) validate this distinction. Since these datasets are programmatically generated, the training and test sets share nearly identical distributions. However, Self-Critique still detects contamination effectively, confirming that our method is instance-specific.
> > >
> > > **2. Feasibility under Diverse Prompt Templates**:
> > > Thank you for pointing out this practical concern. While consistent prompt templates could make rollout results more stable, this setup actually benefits passive methods (like PPL) more, as they rely on absolute probability.
> > > In contrast, the meta-prompt used for the Self-Critique response is independent of the training prompt (as verified by suggested meta-instruction ablation). Our method focuses on measuring the difference under active probing, rather than the properties of the initial generation itself. If the training prompts are very different, we agree that all detection methods will be affected. But compared to methods that rely on a single answer, our active probing approach is less affected by these changes.
> > >
> > > ---
> > > Thank you again for your insightful feedback, which is helpful for defining the actual scope of our work. We will add the clarification for problem 1 to the **Discussion** section and the discussion for problem 2 to the **Limitations** section in our revised paper. Although we notice that the reviewer can't edit the review now, we also hope these clarification could address your concern.

---

### Official Review · Reviewer_iu3G · 2025-11-03

**Soundness:** 3
**Presentation:** 3
**Contribution:** 3
**Rating:** 6
**Confidence:** 4

**Summary:**

This paper addresses the problem of detecting data contamination in LLMs that occurs during the RL post-training phase, a critical gap where existing detection methods, designed for pre-training and SFT, fail. It proposes a novel detection method named Self-Critique and a new benchmark called RL-MIA  to simulate and evaluate this specific type of contamination. The idea of the method is to probe for "policy collapse" (memorization) by instructing the model to generate an initial response and then a second, "self-critique" response with a different reasoning path. The method then compares the token-level entropy sequences of both responses. High similarity in their entropy patterns indicates the model is stuck on a memorized path and is therefore likely contaminated. The empirical studies involve evaluating Self-Critique on the new RL-MIA benchmark, which uses math (AIME 2024/2025) and logical reasoning (K&K, SAT) datasets. The method is tested on models like Qwen2.5-7B and DeepSeek-Math-7B, comparing its performance (using AUC) against several baseline detectors (like PPL, Min-K%, and CDD) and testing its robustness across different RL algorithms (PPO, GRPO, DAPO).

**Strengths:**

This is overall a good paper. The motivating problem is novel and well explained. The paper is the first to systematically study data contamination specifically in the RL post-training phase. It clearly explains why this is a new problem: RL optimizes for a reward signal, not likelihood, which makes existing likelihood-based detectors (like PPL) ineffective (as shown in Table 2, where they perform near random guess).
The method is simple, yet tailored to the problem: The Self-Critique method is an intuitive and clever solution. Instead of checking likelihood, it probes for "policy collapse"—a known side-effect of RL training. The method of comparing the entropy of an initial answer and a "self-critique" answer is a simple and effective way to measure this path dependency.
The writing is very clear. The paper clearly formalizes the problem (Section 3.1) and provides a step-by-step algorithm for the Self-Critique method (Section 4.1, Algorithm 1).
Thorough Experimental Validation: The experiments are comprehensive and support the claims.

- Baselines: It compares Self-Critique against a wide range of baselines (PPL, Min-K%, Recall, CDD) and even custom entropy-based baselines (Entropy-Temp, Entropy-Noise).

- Robustness: It shows the method works across multiple models (Qwen2.5-7B, DeepSeek-Math-7B, etc.), diverse datasets (AIME, K&K, SAT), and different RL algorithms (PPO, GRPO, DAPO, as seen in Table 3).

- Ablations: It includes valuable ablations on the effect of Top-K entropy approximation (Table 4) and sampling strategy (Figure 5).

- Dual-Contamination Study: The analysis in Section 5.4 is particularly strong, showing how Self-Critique can isolate RL-phase contamination even when pre-training contamination is also present, with performance improving up to 55%.

**Weaknesses:**

- Result Variability and Lack of Error Bars: There appears to be significant variability in the performance of some methods across the two main models (Table 2).
- Inference Cost: The Self-Critique method requires the model to generate two full responses (initial and critique) and requires token-level probability access to compute entropy for both. This is computationally more expensive than simpler methods, like PPL, which only require one pass.

**Questions:**

- Table 2 shows significant variability in the performance of baseline methods across different models (e.g., Entropy-Noise on the SAT dataset). Could you provide more intuition on why this variability is so high? Can you bootstrap to get an error bar?
- The paper compares methods with different inference costs (e.g., Self-Critique requires 2 generations, PPL requires 1, and CDD may require N). Could you comment on the performance vs. compute trade-off? For instance, how does Self-Critique compare to a baseline like CDD when both are limited to the same computational budget (e.g., 2 samples)?
- The experiments primarily use Qwen and DeepSeek models. To what extent do you believe the results are general? Would you expect similar performance and baseline rankings if the method were replicated on a different model family, such as Llama, which may have different architectural properties and pre-training data?

I will be happy to increase my score if the authors address these questions.

---

> ### Author Response · Authors · 2025-11-22
> **Author Response (Part 1/2)**
>
> We sincerely thank you for your thoughtful feedback and for recognizing the importance of our work in addressing RL post-training contamination. We will address the noted weaknesses and questions below. **We also provide a revised pdf with new content marked in blue.**
>
>
> `W1 & Q1: Table 2 shows significant variability in the performance of baseline methods across different models (e.g., Entropy-Noise on the SAT dataset). Could you provide more intuition on why this variability is so high? Can you bootstrap to get an error bar?`
>
> Regarding the variability of Entropy-Noise, this method works by injecting a random, non-member prefix into the prompt and measuring the resulting entropy change. The high variability stems from how different models react to these out-of-distribution prefixes in the context of logical reasoning (SAT dataset). Some models may robustly ignore the noise, while others may become unstable or hallucinate wildly, leading to erratic entropy shifts that do not consistently correlate with contamination. In contrast, Self-Critique uses a semantically meaningful "critique" instruction, which guides the model into a more predictable state, resulting in more stable detection.
>
> As suggested, we conducted a Bootstrap analysis (with 1,000 resamples) to calculate the mean AUC and 95% Confidence Intervals (CI) for the SAT dataset. As shown in the table below, while baseline performance fluctuates, Self-Critique demonstrates robust performance with confidence intervals that consistently surpass the likelihood-based methods. We have included this discussion in **Appendix G.1** of the revised PDF.
>
> | Method | Qwen2.5-7B-Instruct Mean ± Std [95% CI] | DeepSeek-Math-7B Mean ± Std [95% CI] |
> | :--- | :--- | :--- |
> | **PPL** | 0.4957 ± 0.0599 [0.3802, 0.6256] | 0.6446 ± 0.0585 [0.5243, 0.7535] |
> | **Min-K%** | 0.5010 ± 0.0601 [0.3721, 0.6149] | 0.3501 ± 0.0582 [0.2384, 0.4709] |
> | **Min-K%++** | 0.3123 ± 0.0539 [0.2113, 0.4247] | 0.4933 ± 0.0571 [0.3818, 0.6070] |
> | **Recall** | 0.6191 ± 0.0552 [0.5092, 0.7248] | 0.6224 ± 0.0598 [0.5042, 0.7396] |
> | **CDD** | 0.4686 ± 0.0560 [0.3620, 0.5811] | 0.4992 ± 0.0193 [0.4615, 0.5369] |
> | **Entropy-Temp** | 0.6877 ± 0.0527 [0.5846, 0.7841] | 0.6069 ± 0.0575 [0.4925, 0.7163] |
> | **Entropy-Noise** | 0.7692 ± 0.0501 [0.6687, 0.8644] | 0.4474 ± 0.0575 [0.3317, 0.5589] |
> | **Self-Critique (Ours)** | **0.6682 ± 0.0540 [0.5583, 0.7607]** | **0.6677 ± 0.0569 [0.5548, 0.7703]** |
>
> `W2 & Q2: The paper compares methods with different inference costs... Could you comment on the performance vs. compute trade-off? For instance, how does Self-Critique compare to a baseline like CDD when both are limited to the same computational budget (e.g., 2 samples)?`
>
> This is a very practical consideration. We classify the methods into two categories: passive (PPL, Min-K%) and active (Recall, CDD, Self-Critique). Passive methods are indeed cheaper (1 pass) but, as our results show, they are largely ineffective for RL contamination because the likelihood signal is decoupled from the training objective. Therefore, the additional cost of active probing is necessary to achieve valid detection.
>
> Among active methods, Self-Critique offers the best balance. Recall also needs two passes (original + prefixed) but generally performs worse than Self-Critique (e.g., 0.59 vs 0.70 Avg AUC on Qwen). CDD relies on measuring consistency across many samples; Following the default setting of CDD paper, we used 50 samples to get a stable estimate. If we limit CDD to just two samples to match Self-Critique’s budget, its estimation of the edit distance distribution becomes noisy and unreliable, leading to meaningless results.
>
> Regarding the token-level probability access, while it adds overhead compared to simple text generation, our ablation study in Table 4 demonstrates that using only the top-K (e.g., K=5) probabilities is sufficient for our entropy calculation, which mitigates the computational burden. In summary, Self-Critique offers the highest detection accuracy per unit of compute among methods that actually work for the RL phase. We have included this discussion in **Appendix G.2** of the revised PDF.

---

> ### Author Response · Authors · 2025-11-22
> **Author Response (Part 2/2)**
>
> `Q3: The experiments primarily use Qwen and DeepSeek models. To what extent do you believe the results are general? Would you expect similar performance and baseline rankings if the method were replicated on a different model family, such as Llama...?`
>
> We agree that testing on a different model family strengthens the generalization claims. During the rebuttal, we trained Llama-3-8B-Instruct using the same pipeline on the K&K and SAT datasets. The results, presented below, align perfectly with our findings on Qwen and DeepSeek.
>
> | Method | K&K (F1) | K&K (AUC) | SAT (F1) | SAT (AUC) |
> | :--- | :---: | :---: | :---: | :---: |
> | **PPL** | 0.52 | 0.47 | 0.67 | 0.53 |
> | **Min-K%** | 0.66 | 0.50 | 0.50 | 0.46 |
> | **Min-K%++** | 0.45 | 0.44 | 0.51 | 0.47 |
> | **Recall** | 0.67 | 0.53 | 0.68 | 0.55 |
> | **CDD** | 0.67 | 0.53 | 0.67 | 0.51 |
> | **Entropy-Temp** | 0.68 | 0.56 | 0.67 | 0.53 |
> | **Entropy-Noise** | 0.67 | 0.52 | 0.68 | 0.57 |
> | **Self-Critique (Ours)** | 0.69 | **0.61** | 0.70 | **0.62** |
>
> As seen above, likelihood-based methods (PPL, Min-K%) remain close to random guessing (AUC ~0.50) on Llama-3, confirming that this is a fundamental property of RL post-training rather than a model-specific quirk. Self-Critique again achieves the highest performance. We have included this result in **Appendix B.1** of the revised PDF. Additionally, in response to Reviewer PKgz, we also validate our method on Llama-3 trained with the standard RLHF paradigm, where Self-Critique similarly outperformed baselines. This result is in **Appendix B.3**
>
> -----
>
> Thanks again for your time and for these important points，your feedback has been a great help in strengthening our paper.

---

### Author Response · Authors · 2025-11-26
**Summary of our Contributions and Revisions**

We sincerely thank all reviewers for their constructive feedback and the time devoted to reviewing our work.  Here we summarize the key strengths highlighted by reviewers and the major revisions included in our revised submission.
## Summary of Strengths and Contributions
- The paper addresses a **critical and underexplored** gap, the motivating problem is **novel and well explained**: data contamination detection in the RL post-training phase. (Reviewer iu3G, 5NYt, PKgz)

> *During the rebuttal period, we notice this issue was echoed by Ilya Sutskever in his [podcast](https://www.dwarkesh.com/p/ilya-sutskever-2). He noted that RL training is often "inadvertently" inspired by benchmarks, making models **"a little too single-minded and narrowly focused."** This aligns with our core motivation: RL induces a specific type of policy collapse that requires new detection methods.*

- The Self-Critique method is **simple, reproducible and effective**. It detects data contamination effectively where traditional likelihood-based methods fail. (Reviewer iu3G, 5NYt, PKgz)


- The experiments are **comprehensive with extensive ablation studies**, involving diverse models, datasets, and baselines. (Reviewer iu3G, PKgz)

- Introduces RL-MIA, a **useful benchmark that may benefit future work** in this area. (Reviewer 5NYt)

- The paper is **easy to follow** and is **well written**. (Reviewer PKgz)

## Revisions during the Discussion
We have added new experiments and additional disscusion. In the revised pdf, **new content is marked in blue**.
- **Robustness and Stability (Appendix C.3 & G.1)**: We add an ablation study on template diversity (showing high stability) and a bootstrap analysis to provide error bars, confirming the reliability of our signal. (Response to Reviewer iu3G, 5NYt)
- **Generalizability to Llama-3 and Standard RLHF (Appendix B.1 & B.3)**: To demonstrate robustness across model families and alignment algorithms, we added experiments on Llama-3-8B and standard RLHF methods (PPO, DPO, TDPO, RTO). Self-Critique consistently outperforms baselines in these settings. (Response to Reviewer iu3G, PKgz)
- **Discussion on Inference Cost (Appendix G.2)**: We add a detailed discussion on the compute vs. performance trade-off. Our method offers the highest detection accuracy under the same compute budget (Response to Reviewer iu3G)
------
Thank you again for the valuable comments that helped us improve our paper.
We hope these revisions could address reviewers' concerns, and we are happy to have further discussion to improve our work.

---

### Meta-Review · Area_Chair_JK6U · 2026-01-02

**Summary:**

I am not sure what to put here as an answer to "Provide a summary of the reviewers' concerns that informed your suggested decision for this paper.", when all issues were addressed. Here is a summary of the reviewers concerns:
* Self-Critique not too novel
* concerns about the reward model
* concerns about the key assumptions of entropy path similarity

**Reviewer Concerns:**

To my mind, the concerns of the reviewers were very carefully and comprehensively addressed.

**Reviewer Scores:**

It’s really hard to say how any reviewer would have changed their score if they had taken part more fully in the discussion. Without hearing it from them directly, anything we write here would just be guesswork.

For this paper, the scores were 6,4,6. I might expect that the reviewers would have raised those by one or two on average, yielding an accept.

---

### Decision · Program_Chairs · 2026-01-26

Accept (Poster)